# A Prospective Randomized Trial of Ropivacaine 5 mg with Sufentanil 2.5 μg as a Test Dose for Detecting Epidural and Intrathecal Injection in Obstetric Patients

**DOI:** 10.3390/jcm12010181

**Published:** 2022-12-26

**Authors:** Yue Li, Yannan Li, Chen Yang, Shaoqiang Huang

**Affiliations:** Department of Anesthesiology, Obstetrics and Gynecology Hospital of Fudan University, Shanghai 200090, China

**Keywords:** accidental intrathecal injection, epidural labor analgesia, epidural test dose, sensitivity, specificity, obstetrics

## Abstract

Objectives: Traditional epidural test dose is somewhat unsuited in obstetrics because of potential risk of severe adverse effects when it is accidentally injected into the subarachnoid space. Some hospitals use a proportion of the total dose of epidural labor analgesia as a test dose. The aim of our study was to assess the effectiveness and safety of ropivacaine 5 mg with sufentanil 2.5 μg to detect intrathecal injection. Methods: This prospective randomized study enrolled parturients who had the demand for epidural labor analgesia and randomly divided them into two groups. Then, 5 mL of 0.1% ropivacaine with sufentanil 2.5 μg was injected into the epidural space or the subarachnoid space in the epidural (EP) group and the intrathecal (IT) group, respectively. The ability to detect intrathecal injection and side effects were assessed to work out the effectiveness and safety. Results: For spinal injection, the sensitivity and the specificity of the symptoms of either warmth or numbness or both assessed at 3 min were both 100%, and the observed negative predictive value (NPV) and positive predictive value (PPV) were 100%. All parturients in the IT group and 2.33% of parturients in the EP group had sensory blockade to cold or pinprick (*p* < 0.0001). A total of 77.55% (38 of 49) of parturients in the IT group were found to have a motor block. A 10 min assessment showed the median cephalad cold and pin levels were T8 and T10, respectively, and the median Bromage score was 4 in the IT group. Incidences of adverse effects in both groups were low. Conclusions: Ropivacaine 5 mg with sufentanil 2.5 μg is effective and safe to detect intrathecal injection as an epidural test dose in obstetric patients.

## 1. Introduction

The concept of an epidural test dose was first introduced by Moore in 1981 [1], which recommended the combination of 45 mg lidocaine and 0.015 mg epinephrine as a test dose to determine the correct placement of the epidural catheter. However, the test dose being applied among the general population is somewhat unsuited in obstetrics. Several studies [2,3,4] have reported extensive sensory and motor block or severe hypotension after unexpected spinal block by the epidural test dose, resulting in unplanned airway management and emergency cesarean section. The American Society of Anesthesiologists Closed Claims database [5] showed that almost two-thirds of high blocks resulted from accidental intrathecal injection through a supposed epidural catheter. Therefore, appropriate approaches to precisely ensure that the catheter is intrathecal without severe adverse effects are important in obstetrics. Clinically, many hospitals use lidocaine 30 mg as the test dose for parturients receiving epidural analgesia to provide relatively reliable and rapid evidence of spinal blockade with fewer side effects [6]. Nevertheless, in obstetrics, intrathecal injection of lidocaine 30 mg [7] also has the inconvenient risk of side effects, such as high sensory block level. 

Some institutions have attempted to use therapeutic doses of anesthetics as the epidural test dose to simplify the procedure and ensure the effect takes place as soon as possible [8]. For epidural labor analgesia, low dosage and low concentration of local anesthetics with or without opioids [9] is recommended as the analgesic bolus in order to ensure ambulation function, which can be considered as a test dose. The German guidelines of an expert association [10] declared that if the initial therapeutic epidural dose does not exceed the routinely used test dose, then a test dose is not required. Because lidocaine 30 mg is roughly equivalent to ropivacaine 12 mg [11,12], the full therapeutic dose for labor analgesia in our hospital, which consists of ropivacaine 15 mg with sufentanil 7.5 μg, exceeds the routinely used epidural test dose of lidocaine 30 mg. Taking into consideration the risk of side effects resulting from accidental intrathecal injection of this therapeutic dose, we often apply ropivacaine 5 mg with sufentanil 2.5 μg as an epidural test dose, which will not cause severe problems even when it is injected into the subarachnoid space, and provide the other ropivacaine 10 mg with sufentanil 5 μg after identification of the right position of the epidural catheter. However, the sensitivity or specificity of this dose in detecting intrathecal placement by a series of clinical manifestations is not yet clear. 

Therefore, we performed this study to determine whether ropivacaine 5 mg with sufentanil 2.5 μg is effective and safe enough as an epidural test dose for intrathecal injection in obstetrics.

## 2. Materials and Methods

### 2.1. Study Design and Participants

This prospective randomized controlled trial was approved by the institutional review board of our hospital on 1 June 2020 (IRB#2020-29) and was registered at the Clinical Research Information Service (ChiCTR 2000035963). All participants provided written informed consent for information collection for research, and the privacy rights of human subjects were observed. 

We enrolled parturients who were ASA physical status 2 and presented for vaginal delivery in our hospital with an indication for epidural labor analgesia. Parturients with local anesthetic allergy, diabetes, mental disorder, drug addiction, and contraindication of neuraxial block were excluded in our study. We also excluded those who were not able or willing to cooperate with the treatment and those who had taken sedative drugs or analgesics within 24 h. Participants were then randomly assigned 1:1 to one of two groups, namely, the epidural (EP) group and the intrathecal (IT) group, using a computer-generated randomization number sequence.

### 2.2. Study Procedures

After assessment of fetal heart rate and cervical dilatation, the participants in our study were transferred to the predelivery room for epidural or spinal anesthesia. Monitoring was performed using noninvasive blood pressure, electrocardiography, and pulse oxygen saturation monitor. Then, the participants were positioned lateral decubitus with their knees bent. The neuraxial anesthetic was performed at the L3–4 interspace. In the EP group, a 20-gauge multiport epidural catheter was advanced 4 cm cephalad into the epidural space, and 5 mL of 0.1% ropivacaine with sufentanil 0.5 μg/mL was administered through it after negative aspiration. If the epidural catheter was detected accidentally misplaced into the subarachnoid space, the case was transferred to the IT group and the catheter was retained for spinal analgesia after the assessment was complete. In the IT group, the same test dose of 5 mL of 0.1% ropivacaine with sufentanil 0.5 μg/mL was administered through a 24-gauge needle by a needle-through-needle technique to the subarachnoid space. After intrathecal injection, the epidural catheter was inserted into the epidural space. Then, the placement of the catheter was preliminary tested by catheter aspiration. If the catheter sucked out cerebrospinal fluid in the IT group, we did not need to carry out any further treatment before the assessment was complete. Afterwards, the parturients were asked to transfer in a left-leaning supine position for the assessment. If the time from the administration of the test dose to the posture of the supine position exceeded three minutes due to catheterization difficulty or other reasons, the case was excluded as well. Cases where the epidural catheter was detected to be accidentally misplaced by sucking out blood in the two groups were eliminated. Parturients who had underwent either accidental dural puncture by the epidural needle or complete cervical dilatation during the ensuing assessment were also eliminated.

### 2.3. Blinding Process

To allow proper blinding, three medical workers participated in the procedure of the study. An anesthesiologist who clearly knew about the methods of the study was responsible for the preparation of the test dose and the operation of the epidural or spinal anesthesia. An assistant in the predelivery room was asked to start the clock after the test dose was injected. The assistant noted down the data on heart rate, blood pressure, and pulse oxygen saturation at different time intervals according to the study and called an anesthesia nurse who was waiting outside of the predelivery room three minutes after the injection to help us assess and take record of the subjective symptoms as well as the level of sensory and motor block as a blinded assessor. The anesthesia nurse had been trained in advance about how to make assessment and take records and was ignorant to the methods of the study. When the blinded assessor was called to enter the predelivery room for assessment, the operation had been completed and the parturient had already been transferred to left-leaning supine position.

### 2.4. Outcome Measures

The primary outcome was evidence of symptoms of warmth and numbness within three minutes. The secondary outcomes included evidence of sensory and motor blockade, the most cephalad sensory level, and side effects related to the test dose.

Clinically, three minutes and five minutes are both acceptable time intervals for the test dose to detect malposition of the epidural catheter [6,13,14]. Hence, subjective symptoms of warmth and numbness in the lower extremities, sensory block to cold and pinprick, and the Bromage score for motor block were assessed three minutes after the administration of the test dose in our study. Because the block level is usually fixed about 10 min after the intrathecal injection of ropivacaine, we assessed the most cephalad cold and pin levels and the Bromage score again in 10 min intervals to mainly observe the safety of the injection of the test dose into the subarachnoid space and the impact of it on ambulation ability. 

The sensory block to cold or pinprick was assessed by alcohol swab or by asking the participant if a plastic needle felt sharp. This sensory assessment started at S1 and moved cephalad until the sensory level could be identified. A high sensory block, which was defined as a sensory level of cold or pinprick above T6, was recorded as a side effect. We used the modified Bromage scale [15] to evaluate the degree of motor block (1 = complete block; 2 = almost complete block; 3 = partial block; 4 = detectable weakness of hip flexion with full flexion of knees while supine; 5 = no detectable weakness of hip flexion while supine; 6 = able to perform partial knee bend (not performed for this study)). A deep motor block (Bromage score ≤2) was also recorded as an adverse effect. Moreover, the heart rate, blood pressure, and pulse oxygen saturation were monitored at 3, 5, and 10 min after the test dose to detect bradycardia, hypotension, and hypoxemia, respectively. Bradycardia and hypoxemia were defined as heart rate below 60 bpm, and pulse oxygen saturation was defined as below 90%. Hypotension was defined as systolic blood pressure below 90 mmHg or a fall of more than 20% compared to the baseline value. After the assessments were completed, the remaining therapeutic dose of ropivacaine 10 mg with sufentanil was given to the EP group, and the epidural catheter was linked to the patient-controlled analgesic pump to provide epidural labor analgesia. If adverse events, such as local anesthetic intoxication, extensive sensory and motor block (sensory block level above T4 with symptoms of chest stuffiness and numbness in the upper extremities with Bromage score of 1), or even total spinal anesthesia, were detected afterwards during the process of routine epidural labor analgesia, the case was recorded in detail as a severe adverse effect.

### 2.5. Statistical Analysis

Because the primary outcome was evidence of symptoms of warmth and numbness in the lower extremities three minutes after the administration of the test dose for detecting intrathecal injection, calculation of the sample size was based on the hypothesis of the proportion of parturients with intrathecal injection in the IT group. On the basis of the supposed 100% detection rate and given a previous study [7] on the sensitivity (100%) and specificity (74%) of the standard lidocaine test dose in obstetrics with a type I error (α) of 0.05 and power of 80% had the total sample size of 82, we determined the required sample size per group as at least 41. Taking into consideration accidental intravascular cannulation rate of 5% [16] and the possibility of precipitate labor, we planned to recruit 100 parturients with 50 parturients in each group. 

Based on an intrathecal catheterization rate of 1:380 to 1:1000 [17,18] after negative aspiration, we used 1:380 to conservatively calculate the negative predictive value (NPV) and the positive predictive value (PPV). A conventional two-by-two table was constructed to determine the sensitivity and specificity of the test for intrathecal injection. Sensitivity is the ability of a test to categorize an individual as “diseased”, while specificity is the ability of a test to categorize an individual as “disease-free”. Sensitivity = true positive/(true positive + false negative); specificity = true negative/(true negative + false positive). Positive predictive value (PPV), negative predictive value (NPV), and likelihood ratio of the test were also calculated using standard formulas. Diagnostic parameters are given with their 95% confidence intervals (95% CI). We used the Shapiro–Wilk test to assess the normality of data. Two-sided *t*-test was used for normally distributed variables, and the Mann–Whitney test was used for variables that were not normally distributed. Fisher exact test was used to compare incidences in our study. Sample size was calculated using PASS version 15. Statistical analysis was performed using SPSS 20.0 (SPSS, Inc., Chicago, IL, USA). *p* < 0.05 was considered statistically significant. 

## 3. Results

From August 2020 to December 2020, a total of 100 parturients were included according to the enrollment criteria, with 50 parturients in each group. No patient in the EP group converted to the IT group because of accidental spinal catheterization. After eliminating cases that had undergone accidental dural puncture (*n* = 1), complete cervical dilatation during the assessment (*n* = 1), or vascular impairment during catheter placement (*n* = 6), 49 parturients in the IT group and 43 parturients in the EP group were enrolled in the final analysis (Table 1). Demographic characteristics, including the maternal age, height, weight, and gestational weeks, did not demonstrate statistical significance between the IT group and the EP group (*p* > 0.05) (Table 1).

All parturients in the IT group reported either warmth or numbness or both in their lower extremity three minutes after intrathecal injection, whereas none of the parturients described these changes in the EP group (*p* < 0.0001) (Table 2). In the three-minute assessment, all parturients in the IT group (49 of 49) and 2.33% (1 of 43) of parturients in the EP group had sensory blockade to cold or pinprick (*p* < 0.0001). One parturient in the EP group reported a blurry sensory change of pinprick on her feet. No parturient in the EP group had a motor block, while 77.55% (38 of 49) of parturients in the IT group were found to have a motor block. In the 10 min assessment, the median level of the cephalad cold and pinprick level were T8 and T10, respectively, in the IT group. Two parturients in the IT group had cephalad sensory levels above T6. The parturient with blurry sensory change of pinprick during the earlier assessment in the EP group showed a slight cephalad pin level of S1. A total of 77.55% (38 of 49) of parturients in the IT group and no parturient in the EP group showed detectable motor block at 10 min. The median Bromage score in the IT group was 4 (detectable weakness of hip flexion with full flexion of knees while supine).

The sensitivity, specificity, and predictive values of the symptoms of warmth and numbness on the lower extremities and the assessment of sensory level and motor block for detecting intrathecal injection of the test dose at three minutes are shown in Table 3. For symptoms of either warmth or numbness or both assessed at 3 min, the sensitivity and specificity were both 100% (95% CI, 92.75%–100% and 91.78%–100%, respectively). According to a presumed negative aspiration intrathecal catheterization of 1:380, the observed NPV and PPV were 100% for the assessment of changes in subjective symptoms (95% CI, 91.78%–100% and 92.75%–100%, respectively). The sensitivity and specificity of the assessment of sensory level were 100% (95% CI, 92.75%–100%) and 97.67% (95% CI, 87.71%–99.94%), respectively. However, the PPV was only 10.17%, reflecting a high false-positive rate. The sensitivity of the assessment of motor blockage was 77.55% (95% CI, 63.38%–88.23%), which was lower than the sensitivity of the sensory assessment.

There was a low incidence of adverse effects in both groups (Table 4). Ten minutes after the test dose, 10.20% (5 of 49) of parturients in the IT group and 2.33% (1 of 43) of parturients in the EP group were found to have hypotension, and 6.12% (3 of 49) of parturients in the IT group reported a high sensory level (≥T6). No parturient in either group had a deep motor block (Bromage score ≤2).

## 4. Discussion

There is no consensus yet on what is a suitable epidural test dose in obstetrics and whether we should still use it. A survey of UK practice in 2005 [19] on obstetric epidural test doses showed astonishing differences in clinical practice. A certain number of hospitals use a proportion of the total dose of epidural labor analgesia as a test dose, which is somewhat similar to the practice in our study.

We chose to use 5 mL of a mixed solution of ropivacaine 5 mg and sufentanil 2.5 μg as the test dose and provided the other 10 mL of ropivacaine 10 mg with sufentanil 5 μg after the identification of the right position of the epidural catheter. Sometimes a dose of ropivacaine 10 mg with sufentanil could also be effective for labor analgesia as an initial therapeutic dose, and it does not exceed the routinely used test dose of lidocaine 30 mg. It seems to be a practical option to administer this whole dose instead of a test dose. However, the initial doses for labor analgesia of many institutions are relatively high in order to provide faster onset time and more satisfactory analgesic effect than low doses. An article in 2016 [20] showed that the initial therapeutic dose for labor analgesia in IWK Health Centre and Stanford University were 10 mL of 0.2% ropivacaine with fentanyl 100 μg and 15 mL of 0.125% bupivacaine with sufentanil 10 μg, respectively, which were much higher than the test dose we routinely use. A recent study in 2021 [21] compared the median effective dose of ropivacaine in patients having epidural labor analgesia during the day and night. They used 3 mL of 0.15% ropivacaine (4.5 mg) as the epidural test dose for epidural labor analgesia, and the single therapeutic dose for the first patient in each group was 18 mg (including the test dose). The test dose and the whole therapeutic dose of ropivacaine were close to the doses in our study. Our study demonstrated that one-third of the therapeutic dose of epidural labor analgesia, including ropivacaine 5 mg with sufentanil 2.5 μg, was effective and safe for detecting accidental intrathecal injection in parturients. 

Because the test dose in the IT group was administered directly through a 24-gauge spinal needle, there could have been a subtle loss of the test dose in the subarachnoid space when the spinal needle was pulled out. Taking into account the inapplicability of intrathecal catheterization in labor analgesia as well as the remaining test dose in the epidural catheter when it was injected through the epidural catheter in the EP group, the tiny difference of the dosage between the two groups resulting from different administration pathways could basically be ignored.

Clinically, there is no very objective measurement to determine the position of the epidural catheter using the test dose. The symptoms of misplacement of the catheter into the vascular system, such as dizziness, blurring of vision, and numbness of lips, or evidence of misplacement of the catheter into the subarachnoid space, such as warmth and numbness in the lower extremities and even the subsequent appearance of sensory and motor block, are judged by the subjective feelings of the patient. An ideal test dose should precisely determine the position of the catheter as fast as possible. As a result, we observed the symptom of warmth or numbness in the lower extremities three minutes after the administration of the test dose to evaluate its effectiveness. In our study, all participants in the IT group reported the symptom of warmth or numbness in their lower extremities three minutes after the test dose, while participants in the EP group did not have such kind of clinical manifestation. In light of high NPV and PPV as well as 100% detection rate at only three minutes, these changes could be convincing evidence of intrathecal injection. Moreover, all the parturients in the IT group had sensory blockade to cold or pinprick. Only one parturient in the EP group, who was only 150 cm tall and weighed 46 kg, had a blurry sensory change of pinprick on her feet. The subsequent successful labor analgesia ruled out the possibility of intrathecal catheterization. As a result, this might be attributed to the low height and weight. The sensitivity and specificity of the sensory level were up to 100% and 97.67%, respectively, but the PPV was only 10.17%, which suggested a high false-positive rate. For evidence of motor block, only 77.55% of parturients in the IT group felt a detectable weakness of hip flexion while supine, and the other parturients in the IT group did not detect weakness of hip flexion. Therefore, the sensory and motor block level were not as convincing as the symptom of warmth or numbness to detect intrathecal injection.

In the 10-min assessment, the median cephalad cold and pin levels were T8 and T10, respectively, in the IT group. Only two parturients in the IT group reported cephalad sensory levels of T6 and T4. We also found that the median and minimum Bromage score in the IT group were 4. These all indicated that ropivacaine 5 mg with sufentanil 2.5 μg was safe for parturients even when it was accidently injected into the subarachnoid space. The parturient with a blurry sensory change of pinprick during the earlier assessment showed a slight cephalad pin level of S1. The skin temperature of her feet specially measured afterward was 34.1 °C. Therefore, besides the reason we mentioned above, we were not able to rule out other factors, such as the influence of low skin temperature on her feet. In view of the results of our study, we have to admit that uncertain judgment of the sensory level may have potential risk of confusion.

As labor analgesia is also called ambulatory epidural analgesia, the test dose should best not affect the motor function in the case of accidental intrathecal injection. Some studies have shown that lidocaine 45 mg impaired the motor function [22,23] and intrathecal administration of lidocaine 30 mg decreased the likelihood of ambulation [7]. The minimum Bromage score of parturients in the IT group in our study was four, which meant that some parturients just felt weakness of hip flexion at worst and did not suffer any apparent effect on ambulation in comparison to intrathecal injection of lidocaine 30 mg.

Because pregnancy is related to increasing sensitivity to local anesthetics, the routinely used test dose for nonpregnant individuals may be too much for parturients. In most studies, different doses of lidocaine were used (e.g., 30 [6], 45 [24], and 60 [25] mg) to work out the appropriate test dose in the nonpregnant population. Although these doses seem to be effective, it may be unsafe in the obstetric population. Stephen Pratt et al. [7] found that 83% parturients who accepted intrathecal injection of 30 mg lidocaine had hypotension, 75% parturients had a high sensory level (≥T6), and 79% parturients reported a high motor block (Bromage ≤2). In our study, none of the participants’ systolic blood pressure was below 90 mmHg, and only 10.20% parturients reported hypotension, which was defined as the systolic blood pressure falling by more than 20% in comparison to the baseline value. Because the participants in our study were those who had labor pain, the decrease in the blood pressure in the intrathecal group might have partly been due to the analgesic effect. We also found that 6.12% parturients in the IT group had a high sensory level (≥T6), and no parturient in either group had a high motor block (Bromage ≤2). It is possible that ropivacaine 5 mg with sufentanil 2.5 μg has fewer side effects than lidocaine 30 mg as a test dose. 

This study had several limitations. First, we did not compare ropivacaine 5 mg plus sufentanil 2.5 μg with the routine test dose of lidocaine 30 or 45 mg. We simply compared our results with the findings in other studies that used lidocaine as a test dose. Further study should be pursued to objectively compare the effectiveness and safety between lidocaine and ropivacaine plus sufentanil test dose. Second, a test dose for epidural anesthesia should not only detect accidental intrathecal placement of the epidural catheter but also detect its accidental intravascular placement. We merely discussed the sensitivity and specificity of the test dose for detection of the intrathecal injection. To our knowledge, intravascular injection of ropivacaine 5 mg is insufficient to cause systemic toxicity. Constrained by ethical reasons, it is infeasible to launch clinical human studies to determine a safety threshold of intravascularly applied ropivacaine, especially in obstetrics. However, we found that a randomized, double-blinded study of ropivacaine 15 mg as a test dose [26] did not report systemic toxicity in the group that received intravascular injection. Therefore, we believe that ropivacaine 5 mg with sufentanil 2.5 μg as a test dose is at least safe for accidental intravascular injection, and we can also observe the effect of epidural analgesia to help identify intravascular catheterization [27]. Lastly, motor blockade was not detected in all patients in the IT group. This might have resulted from not only the low dose of ropivacaine but also a subtle loss of the test dose from the subarachnoid space when the spinal needle was pulled out.

In conclusion, ropivacaine 5 mg with sufentanil 2.5 μg is effective and safe to detect intrathecal placement of the epidural catheter as a test dose in obstetric patients having epidural labor analgesia. The symptom of warmth or numbness in the lower extremities within three minutes after administration of the test dose could be convincing evidence for intrathecal injection detection.

## Figures and Tables

**Table 1 jcm-12-00181-t001:** Demographic characteristics of study population.

	IT (*n* = 49)	EP (*n* = 43)	*p* Value *
Age (y)	29.90 ± 4.5	30.09 ± 3.74	0.492
Height (cm)	162.1 ± 5.97	162.2 ± 5.63	0.912
Weight (kg)	70.20 ± 9.76	67.70 ± 9.31	0.137
Gestation weeks	39.12 ± 1.13	39.16 ± 1.15	0.852

Data are mean ± SD. * *p* < 0.05 was considered statistical significance between the IT group and the EP group. y, year; cm, centimeter; kg, kilogram.

**Table 2 jcm-12-00181-t002:** Degree of sensory and motor blockade in each group.

	IT (*n* = 49)	EP (*n* = 43)	*p* Value
Three-minute interval			
Warmth in the lower extremities	46 (93.80%)	0 (0%)	<0.0001 *
Numbness in the lower extremities	48 (97.96%)	0 (0%)	<0.0001 *
Either of the subjective symptoms	49 (100%)	0 (0%)	<0.0001 *
Sensory blockade level ≥S1	49 (100%)	1 (2.33%)	<0.0001 *
Bromage <5	38 (77.55%)	0 (0%)	<0.0001 *
Ten-minute interval			
Most cephalad cold level	T8 (T4–L2)	None (None)	<0.0001 *
Most cephalad pin level	T10 (T6–L2)	None (None–S1)	<0.0001 *
Bromage score	4 (4–5)	5 (5–5)	<0.0001 *

Data are percentage (*n* %) or median (range). * *p* < 0.05 was considered statistical significance between the IT group and the EP group. Bromage score: 1, complete block; 2, almost complete block; 3, partial block; 4, detectable weakness of hip flexion with full flexion of knees while supine; 5, no detectable weakness of hip flexion while supine; 6, able to perform partial knee bend (not performed for this study).

**Table 3 jcm-12-00181-t003:** Sensitivity, specificity, and predictive values for intrathecal placement at 3 min.

	Sensitivity	Specificity	NPV	PPV	LR+	LR–
Subjective Warmth	93.88%	100%	99.98%	100%	∞	0.06
(95%CI)	(83.13%–98.72%)	(91.78%–100%)	(99.95%–100%)	(92.29%–100%)	(∞–10.38)	(0.01–0.18)
Subjective Numbness	97.96%	100%	99.99%	100%	∞	0.02
(95%CI)	(89.15%–99.95%)	(91.78%–100%)	(99.97%–100%)	(92.60%–100%)	(∞–11.14)	(0–0.12)
Either Subjective Symptom	100%	100%	100%	100%	∞	0
(95%CI)	(92.75%–100%)	(91.78%–100%)	(91.78%–100%)	(92.75%–100%)	(∞–11.59)	(0–0.08)
Sensory Block Level	100%	97.67%	100%	10.17%	42.92	0
(95%CI)	(92.75%–100%)	(87.71%–99.94%)	(99.98%–100%)	(7.62%–13.22%)	(7.73–1000)	(0–0.08)
Motor Block(Bromage <5)	77.55%	100%	99.94%	100%	∞	0.22
(95%CI)	(63.38%–88.23%)	(91.78%–100%)	(99.89%–99.97%)	(90.75%–100%)	(∞–771)	(0.12–0.4)

Data are presented as value (95% confidence interval (CI)); NPV, negative predictive value; PPV, positive predictive value; LR+, positive likelihood ratio; LR-, negative likelihood ratio.

**Table 4 jcm-12-00181-t004:** Adverse effects.

	IT (*n* = 49)	EP (*n* = 43)	*p* Value
Hypotension	5 (10.20%)	1 (2.33%)	0.209
Sensory level ≥T6	3 (6.12%)	0 (0%)	0.245
Bromage ≤2	0 (0%)	0 (0%)	1.000

Data are percentage *n* (%).

## Data Availability

Data available on request due to restrictions of privacy. The data presented in this study are available on request from the corresponding author.

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
