# Peer review of "A Prospective Randomized Trial of Ropivacaine 5 mg with Sufentanil 2.5 μg as a Test Dose for Detecting Epidural and Intrathecal Injection in Obstetric Patients"

_jcm, 2022, doi:10.3390/jcm12010181_

Round 1

Reviewer 1 Report

I reviewed your manuscript titled with “A Prospective Randomized Trial of Ropivacaine 5 mg with Sufentanil 2.5μg as a Test Dose for Detecting Epidural and Intrathecal Injection in Obstetric Patients”.

The authors suggested that Ropivacaine 5mg with sufentanil 2.5μg was effective and safe to detect intrathecal injection as an epidural test dose in obstetric patients.

Examination using test dose, which is used to confirm the exact location of epidural anesthesia, can be said to be one of the important techniques. It is an interesting study in that local anesthetics and opioids were used instead of traditional lidocaine and epinephrine mixed drugs to detect intrathecal injection.

However, there are some points that need to be clarified and revised.

1.     I understand the purpose of this study, but to measure the sensitivity, specificity, and accuracy of detecting the occurrence of an event, it must be compared and analyzed with the results of a standard test or procedure. "ropivacaine 5 mg with sufentanil 2.5 μg" shall be compared with "combination with lidocaine and epinephrine" in order to prove its usefulness as a test dosing for detecting an accidental intrathecal injection. Without these comparative results, the results of this study are reasonable to present only the results of the signs and symptoms that may appear when the study drugs are injected with intrathecal route.

2.     Ethically, it should be stated that the study drug is safe to be injected into an intrathecal route. Currently, drugs used for epidural anesthesia are approved not to be used as intrathecal routes. In particular, when used as an intrathecal route, problems can arise due to preservatives used in drugs. Please provide an opinion that the drugs used in this study are not a problem even when used as intrathecal routes.

3.     It is stated that when the catheter is inserted into the subarachnoid space accidentally in the EP group, they are converted to the IT group. In addition, it is stated that patients with accidental dual puncture were excluded from the study. However, the causes and numbers of these excluded from each group are not described. Also, Figure 1 is missing. Please clarify this part.

4.     Please check the standard of the modified Bromage score used to evaluate the deep and high motor block. Also, please make sure that "modified Bromage score <5" in table 2 is correct.

Author Response

Response to Reviewer 1 Comments

Thank you very much for giving us an opportunity to revise our manuscript. We sincerely appreciate the valuable suggestions to improve the quality of our manuscript. Details of the amendments made according to your opinions are as flowing:

Point 1: I understand the purpose of this study, but to measure the sensitivity, specificity, and accuracy of detecting the occurrence of an event, it must be compared and analyzed with the results of a standard test or procedure. "ropivacaine 5 mg with sufentanil 2.5 μg" shall be compared with "combination with lidocaine and epinephrine" in order to prove its usefulness as a test dosing for detecting an accidental intrathecal injection. Without these comparative results, the results of this study are reasonable to present only the results of the signs and symptoms that may appear when the study drugs are injected with intrathecal route.

Response 1: Thank you for your valuable comment. It is very kind of you to spend your time to read our manuscript with patience and give us an opportunity to improve the quality of our manuscript. Please allow me to offer an explanation to this point. Firstly, some previous studies on standard test dose also used the combination of subjective feelings and objective signs after administration as the judgment basis (Anesth Analg 2013; 116:125-32, Can J Anaesth 2019 Apr;66(4):380-387). Therefore, we think it is reasonable to present these symptoms and signs as the diagnostic basis. Secondly, the diagnostic test may not have to be compared with the standard method. The most important intention is to observe the accuracy of its judgment of the actual situation. What we designed was simulations of different clinical scenes (epidural administration, subarachnoid administration). The symptoms and signs that appeared after the test dose was administered were assessed to judge the actual situation, and thus calculating the sensitivity, specificity and other indicators. Thirdly, the reason why we did not compared the modified test dose with the traditional test dose is that quite a few studies have indicated that the traditionally used test dose is not suitable for obstetrics. The traditional test dose will cause a significant impact on the parturients undergoing labor analgesia (the participants in this study),such as hypotension and obvious motor block when it was injected into the subarachnoid space( in Anesth Analg 2013;116:125-32, the incidences of hypotension and obvious motor block were almost 80%). This may be inappropriate ethically, and may be difficult to pass the ethical approval.

Point 2.     Ethically, it should be stated that the study drug is safe to be injected into an intrathecal route. Currently, drugs used for epidural anesthesia are approved not to be used as intrathecal routes. In particular, when used as an intrathecal route, problems can arise due to preservatives used in drugs. Please provide an opinion that the drugs used in this study are not a problem even when used as intrathecal routes.

Response 2: Thank you for your valuable comment. The drugs we used as a test dose in our study were ropivacaine and sufentanil. Ropivacaine are commonly used intrathecally. Although sufentanil is not routinely used intrathecally in our clinical practice, there are several studied which injected sufentanil into the subarachnoid space in obstetrics (PMID: 27464978 sufentanil 5μg; PMID: 32944556 sufentanil 7.5μg; PMID: PMID: 34338864 meta-analysis). The dose of sufentanil (2.5μg) in our study did not exceed the doses used above. As a result, the study was approved by the institutional review board from our hospital.

According to your requirement, we have provided an opinion that the drugs used in this study are not a problem even when used as intrathecal routes in the introduction section “Taking the risk of side effects resulted from accidently intrathecal injectionspinal injection of this therapeutic dose into consideration, we often apply ropivacaine 5mg with sufentanil 2.5μg as an epidural test dose, which would not bring severe problems even when it was injected into the subarachnoid space”.

Point 3.     It is stated that when the catheter is inserted into the subarachnoid space accidentally in the EP group, they are converted to the IT group. In addition, it is stated that patients with accidental dual puncture were excluded from the study. However, the causes and numbers of these excluded from each group are not described. Also, Figure 1 is missing. Please clarify this part.

Response 3: Thank you for your valuable comment.

If the epidural needle accidentally puncture the dura mater, we normally withdraw the needle and catheterize the epidural catheter into the epidural space. The hole on the dura mater would interfere with the study, so these cases were excluded from the study. If the catheter accidentally entered the subarachnoid space, such cases were conerted to the IT group instead of excluded form the trial.

Actually, no patient in the EP group converted to the IT group because of accidentally spinal catheterization. According to your comment, we have revised this part to “No patient in the EP group converted to the IT group because of accidentally spinal catheterization. After eliminating cases which had undergone either accidental dural puncture (n=1), complete cervical dilatation during the assessment (n=1), or vascular impairment during catheter placement (n=6), 49 parturients in the IT group and 43 parturients in the EP group were enrolled in the final analysis (Figure. 1).” and supplement the CONSORT trial flow diagram (Figure 1) in the manuscript.

Point 4.     Please check the standard of the modified Bromage score used to evaluate the deep and high motor block. Also, please make sure that "modified Bromage score <5" in table 2 is correct.

Response 4: Thank you for your valuable comment. According to your suggestion, we have checked the standard of the modified Bromage score. We found that there are two standards of the modified Bromage scale. Some studies used the modified bromage classification (grade 0: No motor nerve block; 1: Cannot lift leg; 2: Cannot bend knee; 3: Cannot bend the ankle) which is not that applicable in our study because most of the parturients in our study had no motor nerve block or just felt weakness of hip flexion during the period of labor analgesia. As a resulted, we used the modified Bromage scale which is consisted of six grades (1= Complete block; 2=Almost complete block; 3= Partial block; 4= Detectable weakness of hip flexion with full flexion of knees while supine; 5 = No detectable weakness of hip flexion while supine; 6= Able to perform partial knee bend). This modified Bromage scale was also used in other studies (PMID: 23223105 Anesth Analg; PMID: 31331592 Br J Anaesth). The 6th grade “able to perform partial knee bend” was not performed for our study, so we did not mention it in our manuscript at that time. “modified Bromage score <5” in table 2 was to distinguish the parturients with symptoms of motor block.

According to your comment, we have revised the standard of the modified Bromage score to “1= Complete block; 2=Almost complete block; 3= Partial block; 4= Detectable weakness of hip flexion with full flexion of knees while supine; 5 = No detectable weakness of hip flexion while supine; 6= Able to perform partial knee bend (not performed for this study)”, and we have made sure that “modified Bromage score <5” in table 2 is correct. In addition, we have updated the reference about the modified bromage scale to a newer study thanks to your mention (Gabriel, L., Young, J., Hoesli, I., Girard, T., & Dell-Kuster, S. (2019). Generalisability of randomised trials of the programmed intermittent epidural bolus technique for maintenance of labour analgesia: a prospective single centre cohort study. Br J Anaesth, 123(2), e434-e441. doi:10.1016/j.bja.2019.02.016.).

Special thanks to you for your good comments.

Thank you for your time and hope meet your approval.

Best regards

Reviewer 2 Report

Dear Authors,

  My congratulations on a successfully conducted trial. 

The current manuscript is sufficiently well written. The design is clear.   -What was the baricity of local anesthetic? Could the distribution of local anesthetics be affected by the body position?

My main concern is that this test dose was not compared with the standard test dose. How the dose of local anesthetic was chosen? Why exactly 5 mg? How it can be better than 6 or 7 mg?  

Probably with the same success, one might try Ropivacaine 4, 6, 7 mg  and so on. 

This study probably was underpowered to detect severe complications, such as total spinal anesthesia or high block with respiratory compromise or cardiovascular collapse. Since these complications are rare after even a test dose. Although it was an objective of this study.      

Author Response

Thank you for your valuable comments. It is very kind of you to spend your time to read our manuscript with patience and give us an opportunity to revise our manuscript.

Comment1:The current manuscript is sufficiently well written. The design is clear.   -What was the baricity of local anesthetic? Could the distribution of local anesthetics be affected by the body position?

Response 1: Thank you for your valuable comment. The test dose used in our study was Isobaric local anesthetic. As a result, the body position could not affect the distribution of the local anesthetic.

Comment 2:My main concern is that this test dose was not compared with the standard test dose. How the dose of local anesthetic was chosen? Why exactly 5 mg? How it can be better than 6 or 7 mg? Probably with the same success, one might try Ropivacaine 4, 6, 7 mg  and so on. 

Response 2: Thank you for your comment. It was unfortunately one of our limitations in the study design that this test dose was not compared with the standard test dose. There are some reasons for this point. Firstly, some previous studies on standard test dose also used the combination of subjective feelings and objective signs after administration as the judgment basis (Anesth Analg 2013; 116:125-32, Can J Anaesth 2019 Apr;66(4):380-387). Therefore, we think it is reasonable to present these symptoms and signs as the diagnostic basis. Secondly, the diagnostic test may not have to be compared with the standard method. The most important intention is to observe the accuracy of its judgment of the actual situation. What we designed was simulations of different clinical scenes (epidural administration, subarachnoid administration). The symptoms and signs that appeared after the test dose was administered were assessed to judge the actual situation, and thus calculating the sensitivity, specificity and other indicators. Thirdly, the reason why we did not compared the modified test dose with the traditional test dose is that quite a few studies have indicated that the traditionally used test dose is not suitable for obstetrics. The traditional test dose will cause a significant impact on the parturients undergoing labor analgesia (the participants in this study),such as hypotension and obvious motor block when it was injected into the subarachnoid space( in Anesth Analg 2013;116:125-32, the incidences of hypotension and obvious motor block were almost 80%). This may be inappropriate ethically, and may be difficult to pass the ethical approval.

Hence, we were prudent to choose the dose of local anesthetic as a test dose in our study. We found that a recent study in 2021(PMID: 34869403) used 3 ml of ropivacaine 0.15% (4.5 mg) as the epidural test dose for epidural labor analgesia. The concentration of the solution for labor analgesia in many hospitals including our hospital is ropivacaine 0.1%. Therefore, we think 5ml of ropivacaine 0.1% (5mg) is both safe and effective as a test dose, which is easy to be prepared in clinical practice and similar to the dose of the test dose used in the recent study. We did not compare different doses of ropivacaine, so we cannot answer whether 5mg is superior to other doses.The purpose of our research is just to prove that 5mg ropivacaine is both safe and effective as a test dose. The test dose can help us to quickly determine the position of the catheter through its symptoms and signs of blocking without significant adverse effects on the parturients.  

Thank you for your valuable suggestion. Further studies can be launched to explore the minimum effective dose of ropivacaine as the test dose.

Comment 3: This study probably was underpowered to detect severe complications, such as total spinal anesthesia or high block with respiratory compromise or cardiovascular collapse. Since these complications are rare after even a test dose. Although it was an objective of this study.    

Response 3: Thank you for your valuable comment. No case with severe complications was found after the administration of the test dose in both groups even if the test dose was injected into the subarachnoid space in the IT group. In our clinical practice, severe complications such as total spinal anesthesia or high block with respiratory compromise or cardiovascular collapse are more likely to occur when the total therapeutic dose of local anesthetic is accidentally injected into the subarachnoid space without the detection of the test dose. The test dose used in this study (ropivacaine 5mg) was far less than the commonly used dose of ropivacaine (12-15mg) for spinal anesthesia before cesarean section, and in terms of equivalent dose, it was also far less than the standard test dose of lidocaine 30mg (lidocaine 30mg is equivalent to about 12mg of ropivacaine). Therefore, the safety of the modified test dose is higher than the standard test dose. The incidence of hypotension in our study was only about 10% in the IT group, which was far lower than the incidence of about 80% in the other study (Anesth Analg 2013;116:125-32) after the standard test dose was injected into the subarachnoid space. As a result, the risk of more serious complications of the modified test dose is almost unattainable theoretically. We have elaborated on this aspect in the discussion section.

Special thanks to you for your good comments.

Thank you for your time and hope meet your approval.

Best regards

Reviewer 3 Report

The authors present a very interesting clinical trial to evaluate ropivacaine and sufentanil. Please pay attention to the following remarks:

Specify the inclusion criteria.

What is the utility of calculating sensitivity, specificity, and predictive values in this study?

Were there major changes in methods after the start of the trial (such as eligibility criteria), with reasons?

What are the most important limitations of this study?

Could you identify any source of bias?

Could the results of this clinical study improve clinical practice? Why? What is the gold standard treatment in this type of procedure?

Author Response

Thank you very much for giving us an opportunity to revise our manuscript. We sincerely appreciate the valuable suggestions to improve the quality of our manuscript. Details of the amendments made according to your opinions and the response are as flowing:

Comment 1: Specify the inclusion criteria.

Response 1: Thank you for your valuable comment. Since the anesthesia procedure in both groups were suitable for the normally performed labor analgesia, the inclusion criteria in our study was not very tight. There was no restriction on the age, the weight, the parity and so on. We enrolled parturients who presented for vaginal delivery with an indication for epidural labor analgesia. According to your suggestion, we have specified the inclusion criteria in the manuscript that “We enrolled parturients who were ASA physical status 2 and presented for vaginal delivery with an indication for epidural labor analgesia”.

Comment 2: What is the utility of calculating sensitivity, specificity, and predictive values in this study?

Response 2: Thank you for your comment. Sensitivity and specificity are characteristics of a diagnostic test, which need to be considered when deciding whether to adopt this method. Once this method is used to determine the position of the epidural catheter, the significance of the results after administration of the test dose should be carefully considered. What is the possibility of being in the subarachnoid space if the positive result is obtained, and what  is the possibility of not being in the subarachnoid space if the negative result is obtained? These are the predicted values. These characteristics are also conducive to the comparison among different diagnostic tests.

Comment 3: Were there major changes in methods after the start of the trial (such as eligibility criteria), with reasons?

Response 3: Thank you for your comment. In fact, we did not have major changes in our methods after the start of the trial.

Comment 4: What are the most important limitations of this study?

Response 4: Thank you for your good question. We have listed our limitations in the end of the DISCUSSION section, and we think the most important limitation of this study is the first point that we did not compare ropivacaine 5mg plus sufentanil 2.5μg with the routine test dose of lidocaine 30mg or 45mg. We have simply compared our results with the findings in other studies which used lidocaine as a test dose. Further study should be pursued to directly compare the effectiveness and safety between lidocaine and ropivacaine plus sufentanil test dose.

Comment 5: Could you identify any source of bias?

Response 5: Thank you for your comment. The possible bias we can identify is the information bias.

Clinically, there is no very objective measurement to determine the position of the epidural catheter by the use of the epidural test dose. Whether the symptoms of misplacement of the catheter into the vascular such as dizziness, blurring of vision and numbness of lips or the evidences of misplacement of the catheter into the subarachnoid space such as warmth or numbness in the lower extremities and even the subsequent appearance of sensory and motor block are judged by the subjective feelings of the patient. Therefore, it was inevitable that the symptoms of warmth and numbness, and even the sensory and motor block level, which were assessed by the parturients subjectively feelings, may cause information bias to some extent.

In order to decrease the influence of the information bias (including the measurement bias) to the results of the study, we invited three medical workers participated in the blinding process. An anesthesiologist who clearly knew about the methods of the study was responsible for the preparation of the test dose and the operation of the epidural or the spinal anesthesia. An assistant in the predelivery room was asked to start the clock after the test dose was injected. The assistant would note down the data of heart rate, blood pressure and pulse oxygen saturation in different time intervals according to the study and call an anesthesia nurse who was waiting out of the predelivery room three minutes after the injection to help us to assess and take record of the subjective symptoms as well as the level of sensory and motor block as a blinded assessor. The anesthesia nurse had been trained in advance about how to objectively make assessment and take record, and was ignorant to the methods of the study. When the blinded assessor was called to enter the predelivery room for assessment, the operation had been accomplished and the parturient had already transferred in left-leaning supine position. We believe that these measures could minimized the impact of the bias.

Comment 6: Could the results of this clinical study improve clinical practice? Why? What is the gold standard treatment in this type of procedure?

Response 6: Thank you for your valuable remarks. Clinically, the gold standard of detecting the position of the epidural catheter is the lidocaine test dose. However, the test dose being applied among the general population is somewhat unsuited in obstetrics for its higher potential risk of complications such as extensive sensory and motor block or severe hypotension after unexpected spinal block in the obstetric population. The purpose of our study was to assess the effectiveness and safety of a modified test dose which was considered to substitute for the traditional lidocaine test dose in obstetrics especially during epidural labor analgesia. The results showed that the combination solution of sufentanil 2.5μg and ropivacaine 5mg, which was a proportion of the routinely used dosage of labor analgesia, was effective and safe enough to detect spinal injection as a test dose and could be a safer and substitute choice for the lidocaine test dose in parturients with an indication for epidural labor analgesia. Therefore, in my point of view, the results of this clinical study could improve clinical practice.

Special thanks to you for your good comments.

Thank you for your time and hope meet your approval.

Best regards

Reviewer 4 Report

GENERAL REMARK

The authors want to assess the efficacy of a test dose before application of the full dose for epidural analgesia. To determine this they randomize between epidural spinal analgesia and then conclude that in all cases the test dose was effective. However, in spinal analgesia aspiration is common method to test appropriate placement and testing by a test dose is unnecessary. This design does not answer the objective of the study. You should have randomized women for epidural to have a test dose after negative aspiration or only aspiration and compare differences in complications and efficacy of the analgesia.

Furthermore, all complications, like aspiration of liquor by the epidural catheter, or aspiration of blood and failed punctures were excluded, which gives an erroneous presentation of the procedures.

The terms intrathecal, subarachnoid space and spinal are used alternatively – it is better to stick to the more common name of spinal analgesia.

I think this study does not answer the research question and is therefore not very useful. However, I am a gynecologist and not an anesthesiologist, so it might be that a reviewer with this last specialization might have  different opinion.

Author Response

Comment : The authors want to assess the efficacy of a test dose before application of the full dose for epidural analgesia. To determine this they randomize between epidural spinal analgesia and then conclude that in all cases the test dose was effective. However, in spinal analgesia aspiration is common method to test appropriate placement and testing by a test dose is unnecessary. This design does not answer the objective of the study. You should have randomized women for epidural to have a test dose after negative aspiration or only aspiration and compare differences in complications and efficacy of the analgesia.

Furthermore, all complications, like aspiration of liquor by the epidural catheter, or aspiration of blood and failed punctures were excluded, which gives an erroneous presentation of the procedures.

The terms intrathecal, subarachnoid space and spinal are used alternatively – it is better to stick to the more common name of spinal analgesia.

I think this study does not answer the research question and is therefore not very useful. However, I am a gynecologist and not an anesthesiologist, so it might be that a reviewer with this last specialization might have different opinion.

Thank you for your comment. It is very kind of you to spend your time to read our manuscript with patience and give us an opportunity to improve the quality of our manuscript. Please allow me to offer an explanation.

Response: It is correct that aspiration test is commonly used after the placement of the epidural catheter to preliminary estimate the position of the catheter. However, in our clinical practice, negative aspiration does not provide guarantee for correct placement. Anesthesiologists in almost all the hospitals routinely use a test dose (lidocaine or other local anesthetics) after the negative aspiration to further determine the position of the epidural catheter. A study in 2019 (PMID: 30985339) indicated that as negative aspiration tests of epidural catheters provide no guarantee for correct placement, the rationale behind the concept of applying a test dose after catheter insertion is to prevent the potential harm of an either intravenous, intrathecal or subdural injection of a critical amount of local anesthetic or opioid. Taking the potential high risk of complications result from accidentally intrathecal injection of the traditional test dose such as high block with respiratory compromise into consideration, a modified test dose in obstetrics is needed. The purpose of our study was to assess the effectiveness and safety of a modified test dose which was considered to substitute for the traditional lidocaine test dose in obstetrics. The IT group in the study was designed to simulate the scene of accidentally intrathecal misplacement of the epidural catheter. All the parturients in the EP group must be ensured that their epidural catheters were inserted into the epidural space. Therefore, we confirmed it by the successful completion of the subsequent labor analgesia, and the preceding cases of intravenous misplacement (aspiration of blood) and accidental dural puncture were excluded in order to ensure that all the participants were free from uncontrolled factors to eliminate the confounding bias. The cases of aspiration of liquor by the epidural catheter (intrathecal misplacement) were not excluded, but were converted to the IT group because these cases were just suitable for observation of the symptoms and signs of intrathecal injection. In fact, no patient in the EP group converted to the IT group in our study.

Failed punctures were excluded because these cases could not use the test dose via the epidural catheter and thus were impossible to be researched. Actually, no patient in the study underwent failed puncture. We have clarified the process of our study with a CONSORT trials flow diagram (Figure 1) according to your comment. The terms  intrathecal injection, subarachnoid space, spinal analgesia and spinal block conform to the habit of most anesthesiology literature. For example, there are several studies (PMID23223105, PMID30985339, PMID16492853).

Special thanks to you for your good comments. Thank you for your time and hope meet your approval.

Best regards.

Round 2

Reviewer 1 Report

I am very grateful for your best revision of the manuscript and for your sufficient explanation of my comments. I reviewed your manuscript again according to your answer, and I am satisfied with your answer.